**Lake surface-sediment pollen dataset for the alpine meadow vegetation type**
**from the eastern Tibetan Plateau and its potential in past climate reconstructions**
Xianyong Cao[1,2*], Fang Tian[3], Kai Li[4], Jian Ni[4], Xiaoshan Yu[1], Lina Liu[1], Nannan Wang[1]
[1] Alpine Paleoecology and Human Adaptation Group (ALPHA), Key Laboratory of Alpine Ecology, Institute of
Tibetan Plateau Research, Chinese Academy of Sciences, Beijing 100101, China
[2] CAS Center for Excellence in Tibetan Plateau Earth Sciences, Institute of Tibetan Plateau Research, Chinese
Academy of Sciences (CAS), Beijing 100101, China
[3] Beijing Key Laboratory of Resource Environment and GIS, College of Resource Environment and Tourism,
Capital Normal University, Beijing, 100048, China
[4] College of Chemistry and Life Sciences, Zhejiang Normal University, Jinhua, 321004, China
Correspondence: Xianyong Cao (xcao@itpcas.ac.cn)
**Abstract**
A modern pollen dataset with an even distribution of sites is essential for pollen-based
past vegetation and climate estimations. As there were geographical gaps in previous
datasets covering the central and eastern Tibetan Plateau, lake surface-sediment
samples (n=117) were collected from the alpine meadow region on the Tibetan
Plateau between elevations of 3720 and 5170 m a.s.l. Pollen identification and
counting were based on standard approaches, and modern climate data were
interpolated from a robust modern meteorological dataset. A series of numerical
analyses revealed that precipitation is the main climatic determinant of pollen spatial
distribution; Cyperaceae, Ranunculaceae, Rosaceae, and *Salix* indicate wet climatic
conditions, while Poaceae, *Artemisia*, and Chenopodiaceae represent drought. Model
performance of both weighted-averaging partial least squares (WA-PLS) and the
random forest (RF) algorithm suggest that this modern pollen dataset has good



predictive power in estimating the past precipitation for pollen spectra from the
eastern Tibetan Plateau. In addition, a comprehensive modern pollen dataset can be
established by combining our modern pollen dataset with previous datasets, which
will be essential for the reconstruction of vegetation and climatic signals for fossil
pollen sprecta on the Tibetan Plateau. Pollen datasets including both pollen counts
and percentages for each sample together with their site location and climatic data are
available    at    the    National    Tibetan    Plateau    Data    Center    (TPDC;    DOI:
10.11888/Paleoenv.tpdc.271191).
**1 Introduction**
The relationship between modern pollen and climate, and its representation of
vegetation, is the basis for explaining and reconstructing past climate and vegetation
qualitatively or quantitatively (Juggins and Birks, 2012), so improving the quality of
the modern pollen dataset is a primary step for an objective investigation of the
modern relationship and to ensure reliable climate and vegetation reconstructions
(Cao et al., 2018). To make the pollen-source area and taphonomy as compatible as
possible, modern pollen assemblages should be retrieved from the same type of
sedimentary environment as the fossil pollen spectra (Birks et al., 2010). Hence, to
reconstruct past climate and vegetation from fossil pollen extracted from a lacustrine
sediment, a corresponding modern pollen dataset of samples collected from lake
surface-sediments is necessary. Although there are some modern pollen datasets for
the Tibetan Plateau, established to investigate the relationships between pollen and
climate or vegetation (Shen et al., 2006; Herzschuh et al., 2010; Ma et al., 2017), there
are geographical gaps (e.g. the central and eastern Tibetan Plateau) in the sampled
lakes which may bias interpretations.
The available modern pollen datasets reveal that pollen assemblages on the Tibetan
Plateau    are    generally    simple    with    Cyperaceae,    *Artemisia*,    Poaceae,    and
Chenopodiaceae as the dominant taxa (e.g. Herzschuh et al., 2010; Cao et al., 2014),
with arboreal pollen taxa becoming more influential in the marginal areas (e.g Ma et



al., 2017; Li et al., 2020). It is essential to identify the climatic indicators of the
modern pollen taxa (particular for the four dominant taxa) on the Tibetan Plateau,
because the climatic indicators derived from modern pollen datasets from the
surrounding lowland cannot be directly employed on the Tibetan Plateau. With our
current modern pollen dataset extracted from lake surface-sediments we aim to 1) fill
a geographical gap and thus establish a comprehensive modern pollen dataset
covering the entire Tibetan Plateau; 2) determine the climatic indicators for common
pollen taxa from the alpine meadow ecosystem; and 3) evaluate the predictive power
of the modern dataset to reconstruct past climate and assess the reliability of the
random forest algorithm in calibrating the pollen-climate relationship.
**2 Study area**
The elevation range of the lakes sampled for our pollen dataset is between 3720 and
5170 m a.s.l. with a median of 4420 m a.s.l. (the 25% quantile is 4230 m a.s.l and the
75% quantile is 4550 m a.s.l.; Figure 1). Climate of this region is controlled by the
Asian Summer Monsoon in summer with warm and wet climatic conditions, and by
westerlies in winter with cold and dry conditions (Wang, 2006). The eastern and
central Tibetan Plateau containing these sampled lakes (with >4000 m a.s.l elevation)
is covered by alpine meadow with sporadic patches of subalpine shrub. The plant
communities of the alpine medow are dominated by *Kobresia* species (Cyperaceae)
generally, with Ranunculaceae, Asteraceae, *Polygonum* (Polygonaceae), *Potentilla*
(Rosaceae), Fabaceae, and Caryophyllaceae as the common taxa. The subalpine shrub
is gerenally distributed on the northern slopes of mountains with *Salix oritrepha* and
*Potentilla fruticosa* as the main shrub components, while the herbaceous taxa
mentioned above are also common (Wu, 1995; Herzschuh et al., 2010; unpublished
vegetation survey).



**3 Materials and methods**
3.1 Sample collecting and pollen processing
To ensure the even distribution of the representative lakes, we travelled not only along
the hardened roads but also the dirt roads to collect samples from the alpine meadow
on the eastern and central Tibetan Plateau, in July and August 2018. Generally, small
and shallow unnamed lakes (or pools) with less than 100-m radius (n=117) were
selected to reduce the influence of long-distance pollen transported by wind or rivers
(Figure 1). To reduce the influence of the local vegetation component from the lake
shore, the lake surface-sediment samples were collected from the central part of each
lake, with the top 2 cm of lake sediment forming the sample. Although the selected
lakes generally have an even distribution, there is still a gap in the south-west part of
study area because of a lack of road access (Figure 1).

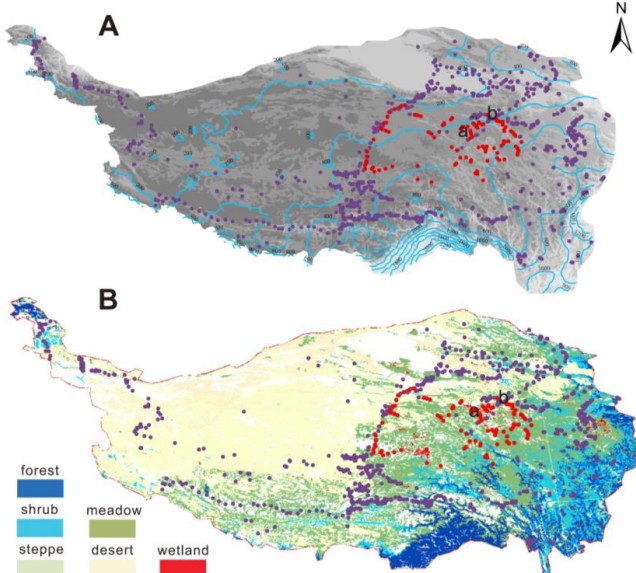


**Figure 1** Spatial distribution of modern pollen samples (red dots: the 117 sampled
lakes; purple dots: previously samples (surface-soils and lake surface-sediments)
included in the dataset of Cao et al., 2014). A: isohyet map (mm); B: vegetation map.
"a" and "b" indicate the locations of Koucha Lake and Xingxinghai Lake.





For pollen extraction, approximately 10 g (wet untreated sediment) per sample were
sub-sampled. Pollen samples were processed using standard acid-alkali-acid
procedures (including 10% HCl, 10% KOH, 40% HF and 9:1 mixture of acetic
anhydride and sulphuric acid successively, Fægri and Iversen, 1975) followed by
7-μm-mesh sieving. A tablet with *Lycopodium* spores (27560 grains/tablet) was added
to each sample prior to pollen extraction as tracers (Maher, 1981). Pollen grains were
identified with the aid of modern pollen reference slides collected from the eastern
and central Tibetan Plateau (including 401 common species of alpine meadow; Cao et
al., 2020) and published atlases for pollen and spores (Wang et al., 1995; Tang et al.,
2017). More than 500 terrestrial pollen grains were counted for each sample.
3.2 Data processing
To obtain modern climatic data for the sampled lakes, the Chinese Meteorological
Forcing Dataset (CMFD; gridded near-surface meteorological dataset) with a
temporal resolution of three hours and a spatial resolution of 0.1° was employed (He
et al., 2020). The CMFD is made through the fusion of remote-sensing products,
reanalysis datasets, and *in situ* station data between January 1979 and December 2018,
and its high reliability has already be confirmed for western China including the
Tibetan Plateau (He et al., 2020). Geographical distances of each sampled lake to each
pixel in the CMFD were calaulated based on their longitude/latitude coordinates using
the *rdist.earth* function in the *fields* package version 9.6.1 (Nychka, et al., 2019) for R
(version 3.6.0; R Core Team, 2019), and the climatic data of the nearest pixel to a
sampled lake were assigned to represent the climatic conditions of that lake. Finally,
the mean annual precipitation ($P_{ann}$; mm), mean annual temperature ($T_{ann}$; °C), and
mean temperature of the coldest month ($Mt_{co}$; °C) and warmest month ($Mt_{wa}$; °C)
were calculated for each sampled lake.
To visualize the relationships between modern pollen assemblages and climatic
variables, ordination techniques were employed based on the square-root transformed
pollen data of 19 taxa (those present in at least 3 samples and with a ≥ 3% maximum)



to stabilize variances and optimize the signal-to-noise ratio (Prentice, 1980).
Detrended correspondence analysis (DCA; Hill and Gauch, 1980) revealed that the
length of the first axis of the pollen data was 1.44 SD (standard deviation units),
indicating a linear response model is suitable for our pollen dataset (ter Braak and
Verdonschot, 1995). We performed redundancy analysis (RDA) to visualize the
distribution of pollen species and sampling sites along the climatic gradients, selecting
the minimal adequate model using forward selection and checking the variance
inflation factors (VIF) at each step. If VIF values were higher than 20, which indicate
that some variables in the model are co-linear, we stopped adding variables (ter Braak
and Prentice, 1988). These ordinations were performed using the *decorana* and *rda*
functions in the *vegan* package version 2.5-4 (Oksanen et al., 2019) for R.
Boosted regression tree (BRT) analysis was applied to determine how strongly the
climatic variables influence the distribution of each individual pollen taxon, using
square-root transformed pollen percentages. A BRT model was generated using the
*gbm.step* function in the *dismo* package 1.0-12 version (Hijmans et al., 2015) for R
with a Gaussian error distribution.
To evaluate the potential of the pollen dataset for past climate reconstruction, both the
traditional method of weighted-averaging partial least squares (WA-PLS) and a new
approach using the random forest (RF) algorithm were run. WA-PLS was performed
using the *WAPLS* function in the *rioja* package version 0.7-3 (Juggins, 2012) for R
using leave-one-out cross-validation, pollen percentages of the 19 selected pollen taxa
were square-root transformed, and the number of WA-PLS components used was
selected using a randomization *t*-test (Juggins and Birks, 2012). We performed the RF
algorithm with the *randomForest* package (version 4.6-14; Liaw, 2018) in R. RF is an
algorithm that integrates multiple decision trees, and the importance of each
explanatory variable is measured as the percentage increase in the residual sum of
squares after randomly shuffling the order of the variables to determine which
explanatory variable can be added to the model. In our study, the importance of all
pollen taxa on the spatial distribution of $P_{ann}$ was estimated and the model



systematically optimized by a stepwise reduction in variables by deleting the least
important one. Our final RF model includes 19 pollen taxa (Appendix 2), which all
make a positive contribution to the precipitation distribution. To assess the predictive
power of our pollen dataset, pollen spectra from Koucha Lake (covering the last 16
cal ka BP; 34.0°N; 97.2°E, 4540 m a.s.l.; Herzschuh et al., 2009; cal ka BP: calibrated
thousand-year before 1950 AD) and Xingxinghai Lake (covering the last 7.5 cal ka
BP; 34.8°N, 98.1°E, 4228 m a.s.l.; Zhang et al., unpublished) were selected as the
target fossil pollen datasets for quantitative reconstruction. A statistical significance
test for all reconstructions was performed following the methods described in Telford
and Birks (2011) using the *randomTF* function in the *palaeoSig* package version 1.1.2
for both WA-PLS and RF reconstruction methods separately (Telford, 2013).
3.3 Data description
Pollen assemblages of the dataset from alpine meadow are dominated by Cyperaceae
(mean 68.4%, maximum 95.9%), with other herbaceous pollen taxa common
including Poaceae (mean 10.3%, maximum 87.7%), Ranunculaceae (mean 4.8%,
maximum 33.6%), *Artemisia* (mean 3.7%, maximum 24.5%), and Asteraceae (mean
2.1%, maximum 33.6%). *Salix* (mean 0.4%, maximum 5.3%) is the major shrub taxon
in these pollen assemblages, while arboreal taxa occur with low percentages generally
(mean total arboreal percentage 0.9%, maximum 5.8%), mainly comprising *Pinus*
(mean 0.3%, maximum 1.8%), *Betula* (mean 0.1%, maximum 0.9%), and *Alnus*
(mean 0.1%, maximum 0.7%). These pollen assemblages represent well the plant
components in the alpine meadow communities, although they are influenced slightly
by long-distance pollen transported by wind or rivers (such as the arboreal pollen taxa;
Figure 2).






**Table 1** Summary statistics for parameters in the pollen dataset. Min.: minimum; Med.: median; Max.: maximum. Units for Longitude and Latitude are degree, for Altitude is m a.s.l., for $Mt_{co}$, $Mt_{wa}$ and $T_{ann}$ are °C, for $P_{ann}$ is mm, while for pollen daxa are %.

| Parameter | Min. | Med. | Max. | Mean | Parameter | Min. | Med. | Max. | Mean |
|---|---|---|---|---|---|---|---|---|---|
| Longitude | 91.80 | 97.20 | 99.79 | 96.42 | *Nitraria* | 0.00 | 0.00 | 0.51 | 0.01 |
| Latitude | 31.59 | 34.02 | 35.52 | 33.74 | Rosaceae | 0.00 | 0.76 | 12.74 | 1.15 |
| Altitude | 3717 | 4422 | 5168 | 4399 | Tamaricaceae | 0.00 | 0.00 | 0.75 | 0.03 |
| $Mt_{co}$ | -19.21 | -15.61 | -7.41 | -15.09 | Apiaceae | 0.00 | 0.16 | 3.98 | 0.32 |
| $Mt_{wa}$ | 3.71 | 6.90 | 11.41 | 7.15 | *Artemisia* | 0.19 | 2.43 | 24.51 | 3.68 |
| $T_{ann}$ | -7.27 | -3.72 | 2.27 | -3.39 | Asteraceae | 0.00 | 1.46 | 33.56 | 2.09 |
| $P_{ann}$ | 226 | 491 | 689 | 471 | Brassicaceae | 0.00 | 0.36 | 28.17 | 1.22 |
| *Abies* | 0.00 | 0.00 | 0.38 | 0.01 | Caryophyllaceae | 0.00 | 0.16 | 2.26 | 0.23 |
| *Cedrus* | 0.00 | 0.00 | 0.19 | 0.00 | Cyperaceae | 4.84 | 76.24 | 95.91 | 68.67 |
| *Picea* | 0.00 | 0.00 | 2.52 | 0.10 | Balsaminaceae | 0.00 | 0.00 | 0.14 | 0.00 |
| *Pinus* | 0.00 | 0.18 | 1.76 | 0.32 | Urticaceae | 0.00 | 0.00 | 3.87 | 0.08 |
| *Alnus* | 0.00 | 0.00 | 0.67 | 0.11 | Gentianaceae | 0.00 | 0.16 | 4.85 | 0.40 |
| *Betula* | 0.00 | 0.00 | 0.94 | 0.11 | Lamiaceae | 0.00 | 0.00 | 1.05 | 0.12 |
| *Carpinus* | 0.00 | 0.00 | 0.63 | 0.06 | Liliaceae | 0.00 | 0.00 | 0.50 | 0.04 |
| *Castanea* | 0.00 | 0.00 | 2.44 | 0.06 | Plantaginaceae | 0.00 | 0.00 | 0.88 | 0.03 |
| *Corylus* | 0.00 | 0.00 | 1.88 | 0.07 | Onagraceae | 0.00 | 0.00 | 0.34 | 0.00 |
| *Juglans* | 0.00 | 0.00 | 0.82 | 0.01 | Papaveraceae | 0.00 | 0.00 | 0.82 | 0.03 |
| Oleaceae | 0.00 | 0.00 | 0.16 | 0.00 | Poaceae | 0.39 | 4.90 | 87.74 | 10.28 |
| *Quercus* | 0.00 | 0.00 | 2.00 | 0.06 | Polemoniaceae | 0.00 | 0.00 | 15.21 | 0.34 |
| *Salix* | 0.00 | 0.18 | 5.35 | 0.45 | *Polygonum* | 0.00 | 0.49 | 20.50 | 1.47 |
| *Ulmus* | 0.00 | 0.00 | 0.16 | 0.00 | *Rumex* | 0.00 | 0.00 | 1.64 | 0.03 |
| Chenopodiaceae | 0.00 | 0.48 | 15.44 | 0.86 | *Koenigia* | 0.00 | 0.00 | 2.96 | 0.39 |
| *Ephedra* | 0.00 | 0.00 | 1.66 | 0.12 | Primulaceae | 0.00 | 0.00 | 0.56 | 0.03 |
| Ericaceae | 0.00 | 0.00 | 0.19 | 0.01 | Ranunculaceae | 0.00 | 3.47 | 33.62 | 4.88 |
| Euphorbiaceae | 0.00 | 0.00 | 0.19 | 0.00 | Saxifragaceae | 0.00 | 0.00 | 4.69 | 0.10 |
| Fabaceae | 0.00 | 0.16 | 3.07 | 0.28 | Scrophulariaceae | 0.00 | 0.00 | 0.71 | 0.01 |
| Hippophae | 0.00 | 0.00 | 5.62 | 0.27 | Solanaceae | 0.00 | 0.00 | 0.69 | 0.01 |
| Rhamnaceae | 0.00 | 0.00 | 0.17 | 0.00 | *Thalictrum* | 0.00 | 0.98 | 12.05 | 1.45 |
| *Ilex* | 0.00 | 0.00 | 0.18 | 0.00 | | | | | |



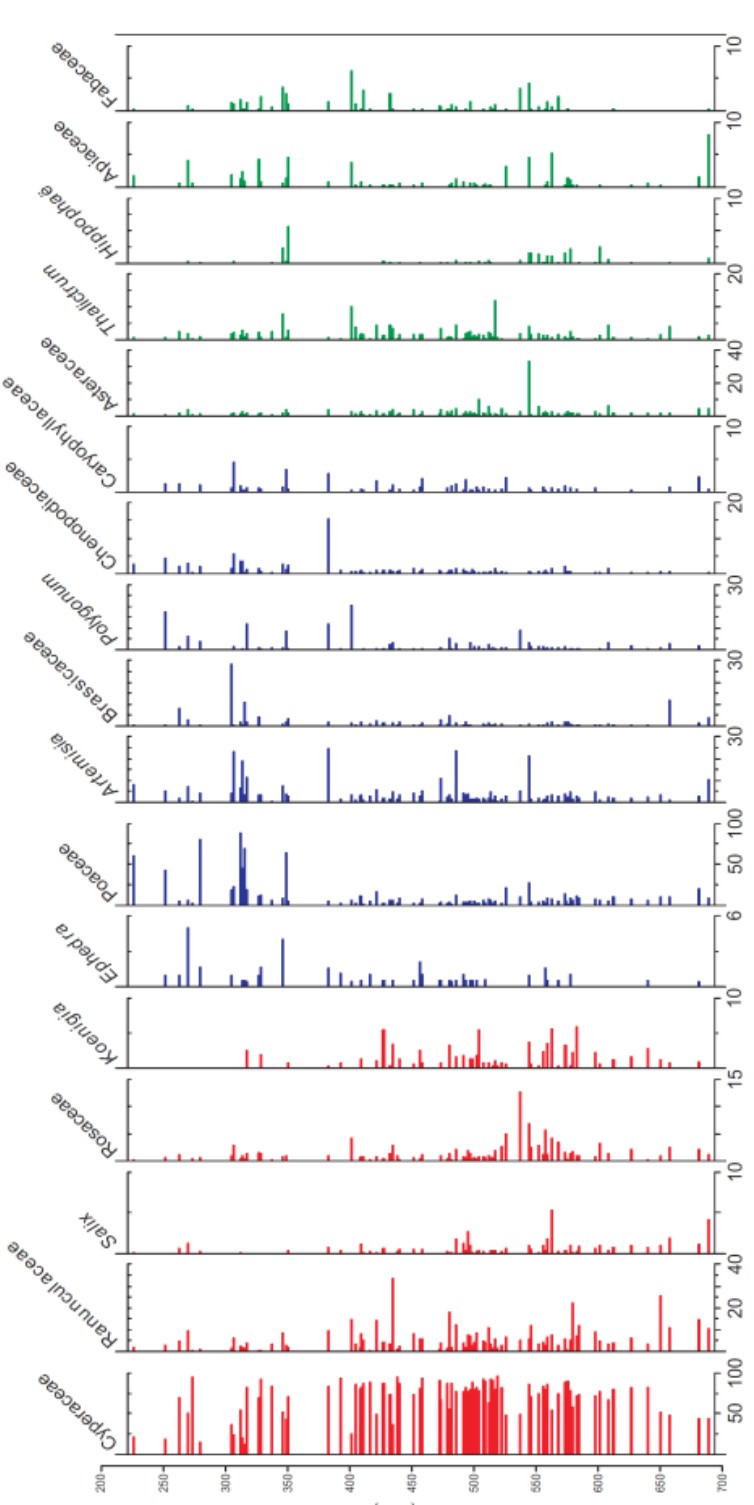

**Figure 2** Pollen diagram showing the major taxa (percentage; %) of the 117 samples arranged by mean annual precipitation ($P_{ann}$; mm).



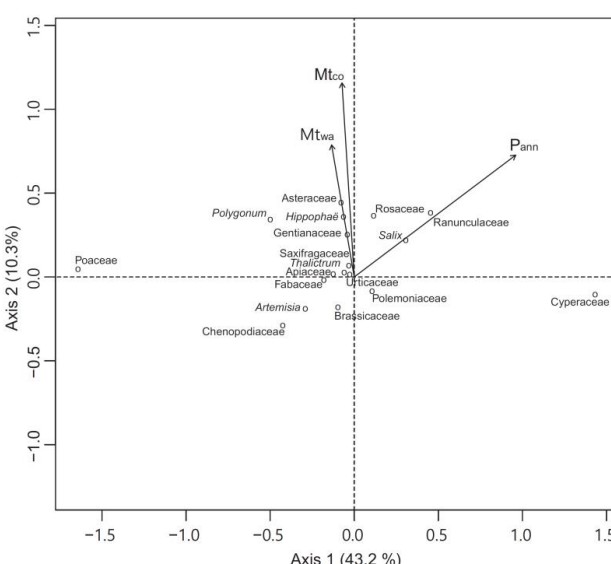

**Figure 3** Plot of the first two redundancy analysis (RDA) axes showing the relationships between 18 pollen taxa (circles) and 3 climatic variables (arrows).

The region covered by these modern pollen samples has a $P_{ann}$ gradient from 226 to 689 mm, and cold thermal conditions with low $T_{ann}$ (-7.3 to 2.3 °C) and $Mt_{co}$ (-19.2 to -7.4 °C). A series of RDAs reveals that, relative to $Mt_{co}$ and $Mt_{wa}$, $P_{ann}$ explains more pollen assemblage variation (10.8% as a sole predictor in RDA) in the dataset (Table 2). A biplot of the RDA shows that the direction of the $P_{ann}$ vector has a smaller angle with the positive direction of Axis 1 (captures 43.2% of total inertia in the dataset) than with the positive direction of Axis 2 (10.3%), indicating that the major component of Axis 1 should be moisture. The RDA separates pollen taxa into two groups generally, Cyperaceae, Ranunculaceae, Rosaceae, and *Salix* indicating wet climatic conditions, while Poaceae, *Artemisia*, and Chenopodiaceae represent drought (Figure 3). Since the low occurrences and abundances for some rare pollen taxa, BRT models are performed successfully for only 14 taxa. BRT modelling results also suggest that $P_{ann}$ is the main climatic determinant for 9 out of 10 of the major pollen taxa with >0.6 prevalence, while Asteraceae is a exception with $Mt_{co}$ as its main climatic determinant (68%; Table 3). BRT results reveal that pollen abundances of



Cyperaceae, Ranunculaceae, and *Salix* are positively relative to $P_{ann}$, while those of
Poaceae, *Artemisia*, and Chenopodiaceae have a negative relationship with $P_{ann}$,
which are consistent with the RDA results (Figure 3 and 4; Appendix 1).
**Table 2** Summary statistics of redundancy analysis (RDA) of 19 pollen species and
four climatic variables. VIF variance inflation factor; $P_{ann}$ annual precipitation (mm);
$Mt_{co}$ mean temperature of the coldest month (℃); $Mt_{wa}$ mean temperature of the
warmest month (℃); $T_{ann}$ annual temperature (℃).

| Climatic variables | VIF (without $T_{ann}$) | VIF (with $T_{ann}$) | Climatic variables as sole predictor | Marginal contribution based on climatic variables | |
|---|---|---|---|---|---|
| | | | Explained variance (%) | Explained variance (%) | $p$-value |
| $P_{ann}$ | 1.6 | 2.9 | 10.8 | 14.7 | 0.001 |
| $Mt_{co}$ | 4.8 | 161.4 | 2.6 | 4.8 | 0.001 |
| $Mt_{wa}$ | 3.8 | 83.9 | 1.6 | 1.3 | 0.100 |
| $T_{ann}$ | - | 447.8 | - | - | - |


**Table 3** Relative influence of climatic variables to the spatial distributions of 14
pollen taxa based on boosted regression tree (BRT) models. For each variable, the
relative influence is expressed as a percentage among the three variables. Pollen taxa
are ordered by decreasing prevalence (the proportion of sites in which each taxon is
present).

| Taxa | Prevalence | $P_{ann}$ | $Mt_{co}$ | $Mt_{wa}$ |
|---|---|---|---|---|
| Cyperaceae | 1.00 | 89.3% | 7.5% | 3.2% |
| Poaceae | 1.00 | 95.1% | 3.3% | 1.5% |
| *Artemisia* | 1.00 | 69.3% | 12.9% | 17.8% |
| Ranunculaceae | 0.99 | 56.9% | 33.7% | 9.4% |
| Asteraceae | 0.97 | 7.2% | 68.0% | 24.8% |
| Rosaceae | 0.90 | 32.2% | 52.7% | 15.1% |
| Chenopodiaceae | 0.85 | 89.1% | 5.8% | 5.1% |
| Brassicaceae | 0.81 | 49.6% | 37.4% | 13.0% |
| *Polygonum* | 0.75 | 42.8% | 31.9% | 25.3% |
| *Salix* | 0.63 | 71.2% | 21.7% | 7.1% |
| Fabaceae | 0.54 | 79.3% | 11.0% | 9.6% |
| Gentianaceae | 0.54 | 10.5% | 63.1% | 26.4% |
| Apiaceae | 0.53 | 33.6% | 30.5% | 35.9% |
| *Hippophaë* | 0.37 | 9.6% | 77.6% | 12.9% |
| Number of > 50% relative influence: | | 7 | 3 | 0 |

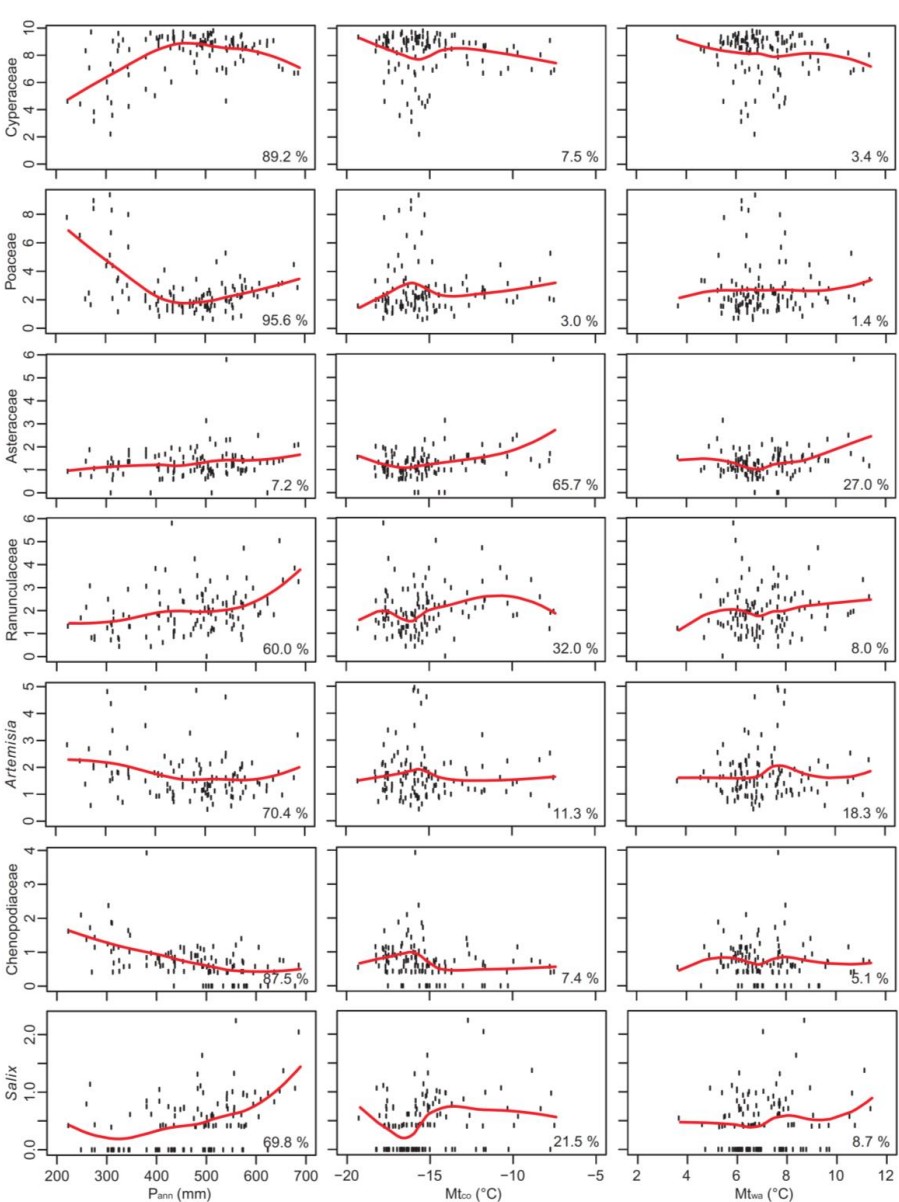

**Figure 4** Boosted regression tree (BRT) modelled climate influences on pollen (seven

dominant or major taxa) percentages. The pollen responses to three climatic variables

(red curves) are fitted with local polynomial regression (LOESS).

## 4 Potential use of the modern pollen dataset

Numerial analyses reveal that $P_{ann}$ is the most important climatic determinant of pollen distribution in the eastern Tibetan Plateau, hence, $P_{ann}$ is selected as the target variable in the calibration-set to assess the predictive power of this pollen dataset. Both approaches (WA-PLS, RF) perform well with low RMSEP values (the root mean square error of prediction) and high $r^2$ values (coefficient of determination between observed and predicted climatic variables; Figure 5). However, the plots of observed vs. predicted $P_{ann}$ show a overestimate of $P_{ann}$ for arid sites and an underestimate for wet sites (Figure 5). Hence, the inevitable "edge effects" should be treated with caution. Nevertheless, the reconstruction with ca. 400–500 mm $P_{ann}$ should be reliable because of the low bias in the central part of the $P_{ann}$ gradient (Figure 5).

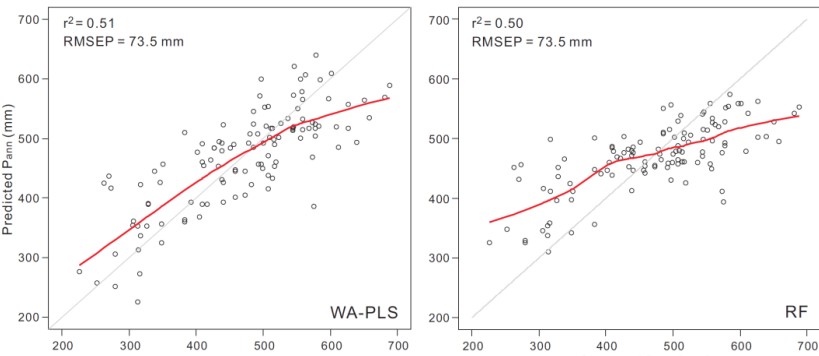

**Figure 5** Scatter plots of observed annual precipitation ($P_{ann}$) vs. predicted $P_{ann}$ by weighted averaging partial least squares regression (WA-PLS) and random forest algorithm (RF).

Although the model performance of RF is not any better than that of WA-PLS, the reconstruction produced by RF might be more reliable as suggested by the statistical significance testing and comparison with modern observed $P_{ann}$ for the two lakes (Koucha Lake and Xingxinghai Lake). Statistical significance testing reveals that reconstructions based on WA-PLS explain less proportion than the 95% quantile of

the proportion of variance explained by random variables (999 times) for the two
lakes, while reconstructions produced by RF explain a higher proportion than the
95% quantile (Figure 6). In other words, reconstructions produced by RF might be
controlled by the major pollen components, because the explained proportion of
variance in the fossil pollen spectra is closer to that explained by the first PCA axis,
while reconstructions by WA-PLS could be influenced more by the pollen taxa with
low abundances (Figure 6). The hypothesis that WA-PLS is more influenced by
low-abundance pollen taxa is supported by the high-variation in reconstructed $P_{ann}$
among the fossil pollen samples (Figure 7). Relative to reconstructions of WA-PLS,
results of RF have lower temporal variation and fewer outliers, and the predicted $P_{ann}$
by RF is closer to the observed $P_{ann}$ for the two lakes (Koucha Lake, 500 mm;
Xingxinghai Lake, 350 mm) than that by WA-PLS.

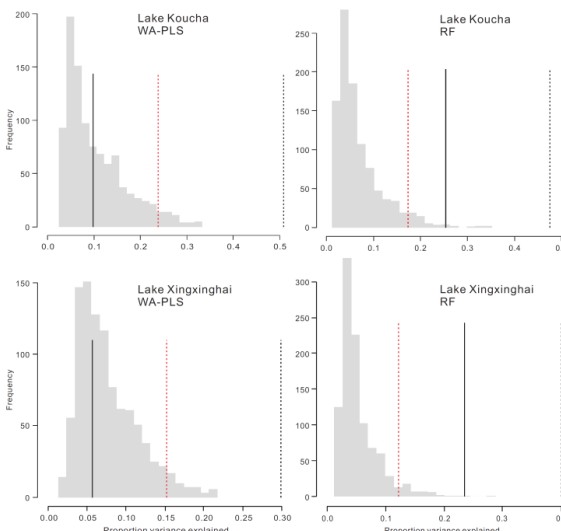


**Figure 6** Statistical significance test of $P_{ann}$ reconstruction from two lakes using
weighted-averaging partial least squares regression (WA-PLS) and the random forest
(RF) algorithm. Grey histograms indicate the proportion of variance in the fossil
pollen spectra explained by random variables (999 times) and the red dotted line is the
95% quantile, the black dotted line is the variance in the pollen explained by the first
PCA axis, and the black solid line is the explanation by the reconstructed $P_{ann}$.

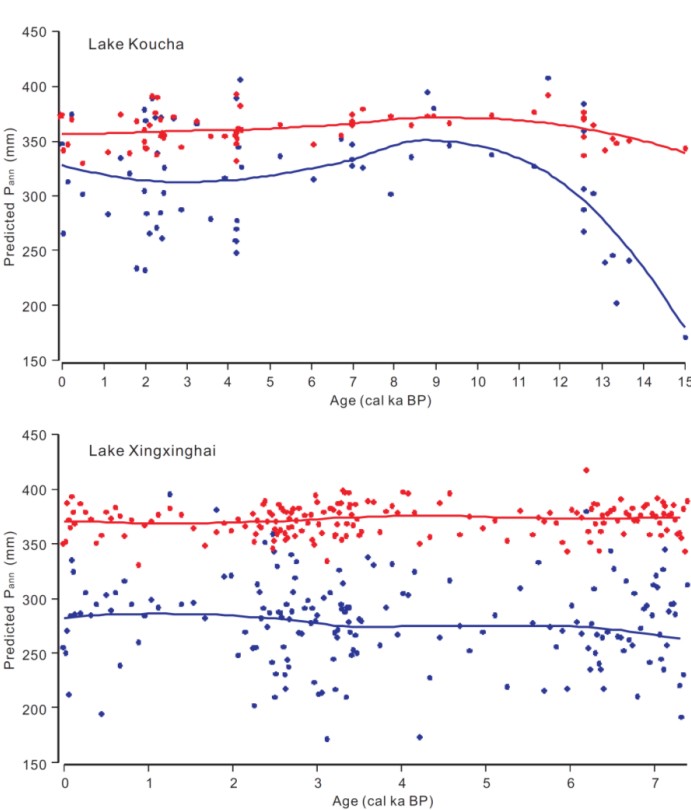


**Figure 7** Annual precipitation ($P_{ann}$; mm) reconstructions for two Tibetan lakes using the weighted-averaging partial least squares regression (blue) and random forest algorithm (red). The curves are fitted by local polynomial regression (LOESS).

## 4 Summary

We present a regional modern pollen dataset extracted from lake surface-sediments from the alpine meadow vegetation type on the Tibetan Plateau (eastern Tibetan Plateau, 91.8°–99.8°E and 31.6°–35.5°N), including pollen counts and pollen percentages together with their positions and climatic data. Numerical analyses reveal that $P_{ann}$ is the most important climatic determinant for pollen distribution in the dataset, and our dataset behaves reliably and has good predictive power for past moisture reconstruction, and the random forest algorithm is a potentially robust approach in pollen-based past environment reconstruction.



In addition, our open-access dataset can fill the geographic gap left by the two
previous modern pollen datasets (lake surface-sediments; Shen et al., 2006;
Herzschuh et al., 2010) on the eastern Tibetan Plateau. By combining our dataset here
with the previous ones (e.g. Herzschuh et al., 2019), a comprehensive modern pollen
dataset is created covering vegetation types from the alpine forest to alpine steppe on
the Tibetan Plateau, and will greatly improve the reliability of past vegetation
reconstructions and climate estimations.
**5 Data availability**
Pollen datasets including both pollen counts and percentages for each sample together
with their locations and climatic data are available at the National Tibetan Plateau
Data Center (TPDC; DOI: 10.11888/Paleoenv.tpdc.271191).
**Author contributions**. XC and JN designed the pollen dataset. XC and KL collected
pollen samples. XY and FT compiled the pollen identification and counting. XC and
FT performed numerical analyses and organized the manuscript, LL and NW prepared
the figures. All authors discussed the results and contributed to the final paper.
**Acknowledgements**
The sample collection and research were supported by the National Natural Science
Foundation of China (Grant No. 41877459 and 41930323), CAS Pioneer Hundred
Talents Program (Xianyong Cao) and Pan-Third Pole Environment Study for a Green
Silk Road of CAS Strategic Priority Research Program (XDA20090000).

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

Appendix A
Boosted regression tree (BRT) modelled climate influences on pollen (seven common
or minor taxa) percentages. The pollen responses to three climatic variableds (red
curves) are fitted with a local polynomial regression (LOESS).

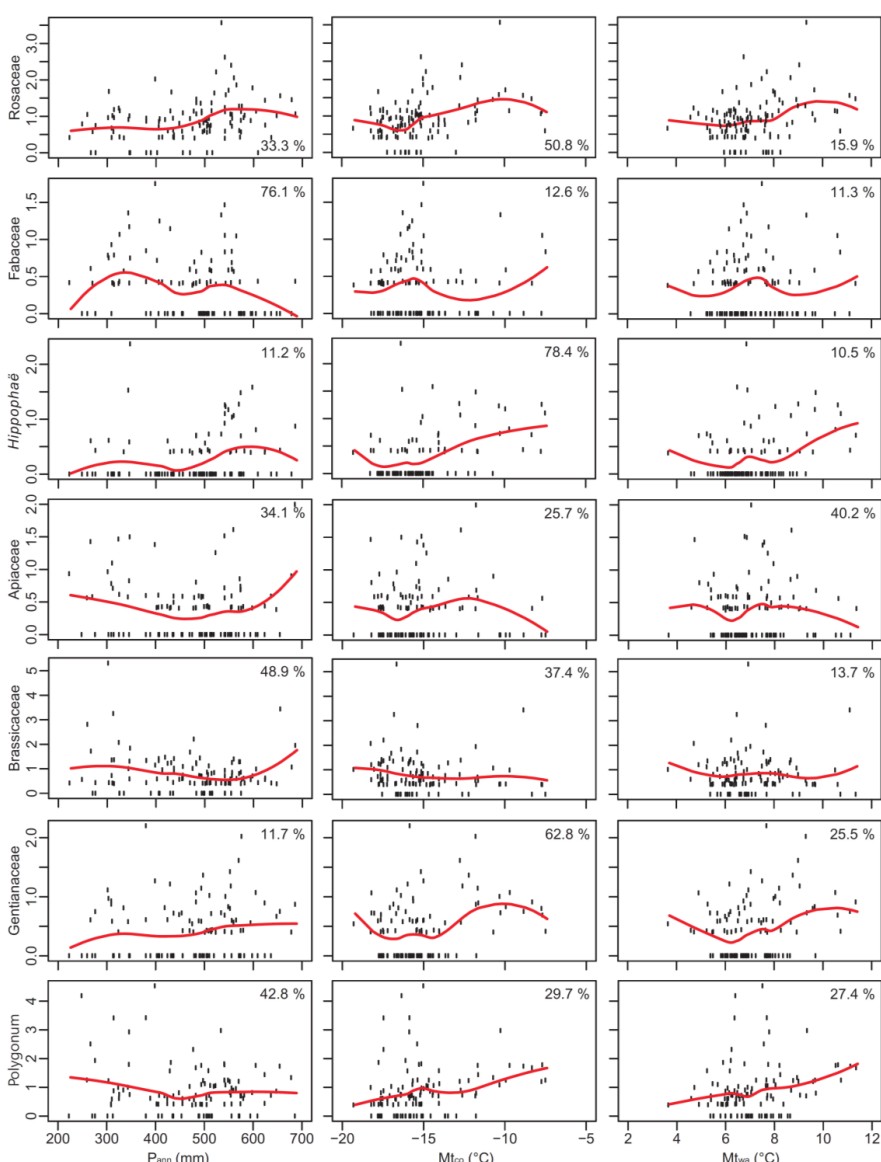







Appendix B Importance (imp) of pollen taxa on the spatial distribution of $P_{ann}$ were
repeatedly assessed by the random forest algorithm (RF). Shown in bold are the
pollen taxa selected for the $P_{ann}$ reconstruction based on RF.

| Taxa | imp-run1 | imp-run2 | imp-run3 | imp-run4 | imp-run5 |
|---|---|---|---|---|---|
| *Abies* | -1.5723 | | | | |
| *Cedrus* | 0.0000 | | | | |
| ***Picea*** | 0.3104 | 3.4397 | 3.5811 | 2.1705 | 1.1599 |
| *Pinus* | -1.6225 | | | | |
| *Alnus* | -0.3501 | | | | |
| ***Betula*** | 5.8217 | 7.4399 | 7.4490 | 5.7763 | 5.9524 |
| *Carpinus* | -1.2049 | | | | |
| *Castanea* | -1.4692 | | | | |
| *Corylus* | 0.2806 | -0.3715 | | | |
| *Juglans* | 0.0000 | | | | |
| Oleaceae | 0.0000 | | | | |
| *Quercus* | -0.4776 | | | | |
| ***Salix*** | 9.2463 | 9.6372 | 10.0018 | 9.4944 | 10.2897 |
| *Ulmus* | -0.6041 | | | | |
| **Chenopodiaceae** | 17.7282 | 18.0369 | 16.8653 | 16.3110 | 18.5089 |
| ***Ephedra*** | 2.8306 | 2.9972 | 4.4539 | 3.5096 | 4.0226 |
| Ericaceae | 0.0755 | 1.7893 | -0.2415 | | |
| Euphorbiaceae | -0.9748 | | | | |
| **Fabaceae** | 2.4847 | 2.5302 | 3.5031 | 3.2985 | 1.8323 |
| ***Hippophaë*** | 5.5569 | 3.5027 | 4.0142 | 3.1174 | 4.5627 |
| Rhamnaceae | 0.0000 | | | | |
| *Ilex* | 0.0000 | | | | |
| *Nitraria* | -1.0010 | | | | |
| **Rosaceae** | 3.0053 | 4.8099 | 2.9771 | 3.6032 | 4.3940 |
| Tamaricaceae | -2.3780 | | | | |
| Apiaceae | -0.6466 | | | | |
| *Artemisia* | 1.7355 | -0.0902 | | | |
| Asteraceae | 2.3902 | 1.7955 | 1.1307 | -1.0880 | |
| **Brassicaceae** | 1.7269 | 2.2776 | 1.4596 | 1.5560 | 1.5308 |
| Caryophyllaceae | -0.0033 | | | | |
| **Cyperaceae** | 9.9824 | 9.8975 | 11.1838 | 10.4553 | 10.3560 |
| Balsaminaceae | 0.0000 | | | | |
| Urticaceae | 0.8534 | -1.4774 | | | |
| Gentianaceae | 1.1305 | -0.8603 | | | |
| **Lamiaceae** | 3.3097 | 2.6853 | 3.4047 | 2.2080 | 2.6588 |
| Liliaceae | -0.5353 | | | | |
| **Plantaginaceae** | 2.3294 | 1.3210 | 1.4498 | 0.8906 | 0.8763 |
| Onagraceae | 1.0010 | -0.8613 | | | |
| Papaveraceae | 0.1148 | 1.0344 | -1.7028 | | |





| | | | | | |
|---|---|---|---|---|---|
| **Poaceae** | 13.8815 | 14.5295 | 14.7793 | 15.7914 | 16.2655 |
| Polemoniaceae | -0.5507 | | | | |
| *Polygonum* | 0.0523 | 2.4552 | 2.9776 | 1.9432 | 2.3618 |
| *Rumex* | 1.0010 | 0.0000 | | | |
| *Koenigia* | 5.4498 | 4.3961 | 3.3305 | 4.1574 | 4.9186 |
| Primulaceae | -1.2283 | | | | |
| **Ranunculaceae** | 6.4799 | 8.9763 | 7.6140 | 7.5498 | 5.5157 |
| **Saxifragaceae** | 0.9422 | 1.3283 | 1.8760 | 4.1134 | 2.3728 |
| Scrophulariaceae | -1.0010 | | | | |
| Solanaceae | 1.0010 | -1.0008 | | | |
| *Thalictrum* | 2.9345 | 2.3850 | 2.6363 | 2.4267 | 3.3457 |
