# Peer review of "Lake surface-sediment pollen dataset for the alpine meadow vegetation type from"

_Earth System Science Data, 2021_

## Referee Comment (RC2)

General comments

This study reported the construction of the modern pollen dataset in eastern and central Tibetan Plateau by collecting samples from 117 lakes and pools between elevations of 3720 and 5170 m a.s.l.. which compensate for the geographical data gap in the region. Precipitation is found to be the main climatic determinant of pollen spatial distribution, by interpolating the pollen dataset against a robust modern meteorological dataset. The capability of the pollen dataset in reconstructing past climate is then tested and verified using using two deep lake sediment cores.

The topic is important for the climate and environmental reconstruction for Tibetan and beyond. However the study, in its current layout, requires certain revision before it could be considered for publication in this journal. My main comments are:

Specific comments
1. Please provide a general framework of the reconstruction analysis. For example, please provide brief justification of how the pollen species of present climate conditions can be used in past climate reconstruction, especially for the Holocene and earlier climate.
2. Regarding "Sample collection": I do not quite understand the criteria of lake selection (Line 83-86) mean exactly. In particular, how does the choice of small and unnamed lakes or pools reduce the influence of long-distance pollen transport? Did the authors mean they reduced the sampling distance by including 117 small lakes and pools?
3. Regarding "Data processing": the authors did not mention the time period of the CMFD data they used for analysis. Does it include the entire period of Jan 1979 to Dec 2018?
4. What time frame does the 2cm surface pollen sample represent? Was it more or less uniform across all sample sites? Does the time frame comparable to the CMFD observation?
5. The information in Figure 2 is difficult to extract, and it is rarely referred to and therefore illustrated in the text.
6. The same applies to Figure 3. The results should be illustrated in more detail. For example, how were the pollens classified as wet-indicator vs. drought-indicator (Line 198-201)? By their locations (positive vs. negative) in Axis 1? What do the two arrows for temperatures indicate?

Technical corrections
1. Please provide information for the color bars in Figure 2.
2. Please still provide the full name of the three climate variables, together with their symbols, in the caption of Figure 3.
3. Please explain why certain pollen's name are in in italic while the others remain normal.
4. Please consider separate the statistics of the geographic and climate parameters from those of the pollen data, either within Table 1, or using two tables.
5. The language could be further polished, for example in Line 201-202, to avoid

possible confusion or ambiguity.

6. Please use mm/yr or mm/a as unit for precipitation.

---

## Author Comment (AC1)

**Response to comments of Anonymous Referee #1**

**Summary**

Cao et al. describe a new database of modern lake sediment pollen samples from the central and eastern Tibetan Plateau. Additionally, the authors utilize two statistical analytical metrics, weighted-averaging partial squares and random forest, in order to determine the climatic controls to pollen distribution. Additional attention is needed in small but key areas including, minor clarifications within the methodology and overall MS structure. Beyond these minor changes, this paper and accompanying dataset will compliment existing pollen datasets on the Tibetan Plateau as well as provide useful data for paleoclimate studies moving forward.

**Main Comments (Manuscript)**

Some of the methodology is insufficient. While I can understand the necessity of road access, I would be interested to see whether a GoogleEarth or similar initial scouting of the area would produce a more even distribution of sample sites. Additionally, there is mention of long-distance pollen transport within the data description section (lines 173-176). However, no explanation how this is determined is given. You mentioned that bodies of water directly connected to rivers were avoided but did not go further. Please clarify this assertion. Within the methodology section, there is no discussion regarding sources of error (sampling, etc.) or how they are treated. Please clarify.

**Response: The Tibetan Plateau is a quite remote area and road access is essential for collecting samples. In addition, the south-west part of the study area is mountainous with few lakes, hence our modern pollen dataset still has a geographical gap in the south-west part. We explain the reason for the uneven distribution in the new version. We also explain how we attempted to minimise the long-distance pollen transport by selecting small lakes. Generally, pollen grains are identified and counted under an optical microscope. To ensure a reliable representation of the entire pollen assemblage by the counted pollen data (reducing the error of pollen analysis), an adequate number of pollen grains and *Lycopodium* spores should be counted for each sample, which we now discuss.**

Line 89-98:

"*To reduce the influence of long-distance pollen transported by wind and rivers, small and shallow lakes (or pools) with less than 100-m radius and without long inflow rivers (n=117) (locally sourced pollen grains are the dominant components for small lakes; Sugita, 1993) were selected to collect pollen samples (Figure 1). To reduce the influence of the lake-shore vegetation component, the lake surface-sediment samples were collected from the central part of each lake, with the top 2 cm of lake sediment forming the sample (Tian et al., 2008). Although the selected lakes generally have an even distribution, there is still a gap in the south-west part of study area because of a lack of lake and road access (Figure 1).*"

Line 108-111:

"*More than 500 terrestrial pollen grains were counted for each sample, and more than 200 Lycopodium spores were counted for most of the samples (mean=270 grains; median=480 grains), both of which ensure a reliable representation of the entire pollen assemblage by the counted pollen data.*"

Next, it would make the MS easier to read if subsection "3.3 Data Description" were its own section (e.g., 4 Results). Finally, while overall the use of English is good, there are minor yet reoccurring instances of run on sentences that should be addressed.

**Response: We modified the subsection "3.3 Data Description" as an independent section "4 Data Description" in the new version. And the written English was polished by a native English speaker.**

**Main Comments (Data)**

The database is easily accessed and downloaded. The database itself is a two page Excel file that clearly presents the individual counts of pollen as well as their overall percentage at each site. Providing both counts and percentages is a good touch. However, adding errors to the sample meta data (Elevation, $Mt_{co}$, $Mt_{wa}$, $T_{ann}$, and $P_{ann}$) would be useful for future users of the dataset.

**Response: The climate data were obtained from the Chinese Meteorological Forcing Dataset (He et al., 2020, Scientific Data) and in the original publication, He et al. describe data quality and error. Hence, we decided not to add that into our manuscript.**

**Minor comments**

line 154: Appendix 2 does not exist, it seems to be referencing Appendix B, please clarify.

**Response: Corrected. "Appendix 2" should be "Appendix B".**

Line 154-155:

"*Our final RF model includes 19 pollen taxa (Appendix B), which all make a positive contribution to the precipitation distribution.*"

Please double check section numbering once structure is fixed.

**Response: Done.**

**Figures:**

Figure 1) Please provide rain fall scale for isohyet map. Make 'a' and 'b' more clearly visible. Also Line 67 references Figure 1, but Figure 1 does not have relevance to elevation discussion in this sentence. A 3rd subfigure in Figure 1 with elevation of the region would help.

**Response: Done. We have modified the two subfigures and add a new one with an elevation map.**

Figure 2) The choice of colors within the figure are not clear, please explain in figure caption.

**Response: Done.**

Page 9:

"***Figure 2*** *Pollen diagram showing the major taxa (percentage; %) of the 117 samples arranged by mean annual precipitation ($P_{ann}$; mm). Pollen with red bars are positively related to $P_{ann}$, those with blue bars are negatively related to $P_{ann}$, while the relationship is insignificant for those with green bars.*"

---

## Author Comment (AC2)

**Response to comments of Anonymous Referee #2**

General comments

This study reported the construction of the modern pollen dataset in eastern and central Tibetan Plateau by collecting samples from 117 lakes and pools between elevations of 3720 and 5170 m a.s.l.. which compensate for the geographical data gap in the region. Precipitation is found to be the main climatic determinant of pollen spatial distribution, by interpolating the pollen dataset against a robust modern meteorological dataset. The capability of the pollen dataset in reconstructing past climate is then tested and verified using two deep lake sediment cores.

The topic is important for the climate and environmental reconstruction for Tibetan and beyond. However the study, in its current layout, requires certain revision before it could be considered for publication in this journal. My main comments are:

Specific comments

**1**. Please provide a general framework of the reconstruction analysis. For example, please provide brief justification of how the pollen species of present climate conditions can be used in past climate reconstruction, especially for the Holocene and earlier climate.

**Response: We agree with this comment, and we have provided a general framework of pollen-based past climate reconstruction in the new version.**

Line 143-147:

*"The basic assumption of pollen-based past climate reconstruction assumes that pollen taxa recorded in the modern calibration-set have similar ecological requirements as those in the fossil spectra (Juggins and Birks, 2012); in other words, the modern vegetation-climate relationship is assumed to be stable temporally through the target period of reconstruction."*

2. Regarding "Sample collection": I do not quite understand the criteria of lake selection (Line 83-86) mean exactly. In particular, how does the choice of small and unnamed lakes or pools reduce the influence of long-distance pollen transport? Did the authors mean they reduced the sampling distance by including 117 small lakes and pools?

**Response: A small lake has a small pollen source area as confirmed by previous experiments and modelling. In the new version, we cite the literature to support our argument.**

Line 89-98:

*"To reduce the influence of long-distance pollen transported by wind and rivers, small and shallow lakes (or pools) with less than 100-m radius and without long inflow rivers (n=117) (locally sourced pollen grains are the dominant components for small lake; Sugita, 1993) were selected to collect pollen samples (Figure 1). To reduce the influence of the lake-shore vegetation component, the lake surface-sediment samples were collected from the central part of each lake, with the top 2 cm of lake sediment forming the sample (Tian et al., 2008). Although the selected lakes generally have an even distribution, there is still a gap in the south-west part of study area because of a lack of lake and road access (Figure 1)."*

3. Regarding "Data processing": the authors did not mention the time period of the CMFD data they used for analysis. Does it include the entire period of Jan 1979 to Dec 2018?

**Response: We describe the meteorological data in more detail in the new version. The CMFD includes the continuous data for the entire period of Jan 1979 to Dec 2018.**

Line 115-124:

*"Geographical distances of each sampled lake to each pixel in the CMFD were calculated based on their longitude/latitude coordinates using the rdist.earth function in the fields package version 9.6.1 (Nychka, et al., 2019) for R (version 3.6.0; R Core Team, 2019), and the meteorological data (three-hour resolution between January 1979 and December 2018) of the nearest pixel to a sampled lake were assigned to represent the climatic conditions of that lake. Finally, the mean annual precipitation ($P_{ann}$; mm), mean annual temperature ($T_{ann}$; °C), and mean temperature of the coldest month ($Mt_{co}$; °C) and warmest month ($Mt_{wa}$; °C) were calculated for each sampled lake based on the long-term continuous meteorological data."*

4. What time frame does the 2-cm surface pollen sample represent? Was it more or less uniform across all sample sites? Does the time frame comparable to the CMFD observation?

**Response: It is quite difficult to ascertain the time frame for the 2-cm surface-sediment. The unpublished Pb/Cs dating results for a series of lakes from the east and central Tibetan Plateau completed by our research group, indicate that the 2-cm surface-sediment covers the last 20~80 years. The CMFD includes continuous data for the entire period of Jan 1979 to Dec 2018, covering the last 40 years.**

5. The information in Figure 2 is difficult to extract, and it is rarely referred to and therefore

illustrated in the text.

**Response: Figure 2 presents the pollen diagram for the 117 samples. From this figure, we can see the pollen-Pann responses for the dominant and common pollen taxa. In the new version, we explain what the different colours mean.**

Page 11:

*"Figure 2 Pollen diagram showing the major taxa (percentage; %) of the 117 samples arranged by mean annual precipitation ($P_{ann}$; mm). Pollen taxa with red bars are positively related to $P_{ann}$, those with blue bars are negatively related to $P_{ann}$, while the relationship is insignificant for those with green bars."*

6. The same applies to Figure 3. The results should be illustrated in more detail. For example, how were the pollens classified as wet-indicator vs. drought-indicator (Line 198-201)? By their locations (positive vs. negative) in Axis 1? What do the two arrows for temperatures indicate?

**Response: Done.**

Line 196-206:

*"A biplot of the RDA shows that the direction of the Pann vector has a smaller angle with the positive direction of axis 1 (captures 43.2% of total inertia in the dataset) than with the positive direction of axis 2 (10.3%), indicating that the major component of axis 1 should be moisture. RDA axis 1, which is highly correlated with Pann, divides pollen taxa into two groups generally: Cyperaceae, Ranunculaceae, Rosaceae, and Salix indicating wet climatic conditions (located along the positive direction of Pann), while Poaceae, Artemisia, and Chenopodiaceae represent drought (located along the negative direction of Pann; Figure 3). Axis 2 is highly correlated with the two temperature variables, however these dominant pollen taxa have insignificant distributions along the axis, hence temperaturee is the secondary climatic variable for the pollen dataset relative to precipitation (Figure 3)."*

**Technical corrections**

1. Please provide information for the color bars in Figure 2.

**Response: Done (see above).**

2. Please still provide the full name of the three climate variables, together with their symbols, in the caption of Figure 3.

**Response: Done.**

Line 212-215:

*"Figure 3 Plot of the first two redundancy analysis (RDA) axes showing the relationships between 18 pollen taxa (circles) and 3 climatic variables (arrows). $P_{ann}$: mean annual precipitation (mm); $Mt_{co}$: mean temperature of the coldest month (°C); $Mt_{wa}$: mean temperature of the warmest month (°C)."*

3. Please explain why certain pollen's name are in italic while the others remain normal.

**Response: Generally, pollen grains can be identified into family, genus or species levels. Latin names for genus- and species-level pollen taxa should be in italic, while those for family-level taxa should be normal. It is the specification for Latin names of animals and plants.**

4. Please consider separate the statistics of the geographic and climate parameters from those of the pollen data, either within Table 1, or using two tables.

**Response: We have separated the statistics of the geographic and climate parameters from those of pollen taxa in the new version of Table 1.**

5. The language could be further polished, for example in Line 201-202, to avoid possible confusion or ambiguity.

**Response: Written English has been polished by a native English speaker again for the new version.**

6. Please use mm/yr or mm/a as unit for precipitation.

**Response: $P_{ann}$ is mean "annual" precipitation (mm), so "mm" is appropriate as the unit.**